# Towards Antibacterial Agents: Synthesis and Biological Activity of Multivalent Amide Derivatives of Thiacalix[4]arene with Hydroxyl and Amine Groups

**DOI:** 10.3390/pharmaceutics15122731

**Published:** 2023-12-05

**Authors:** Igor Shiabiev, Dmitry Pysin, Alan Akhmedov, Olga Babaeva, Vasily Babaev, Anna Lyubina, Alexandra Voloshina, Konstantin Petrov, Pavel Padnya, Ivan Stoikov

**Affiliations:** 1A.M. Butlerov Chemical Institute, Kazan Federal University, Kremlevskaya, 18, Kazan 420008, Russia; shiabiev.ig@yandex.ru (I.S.); pysin_dima@mail.ru (D.P.); naive2294@gmail.com (A.A.); 2Arbuzov Institute of Organic and Physical Chemistry, FRC Kazan Scientific Center, Russian Academy of Sciences, 8 Arbuzov Street, Kazan 420088, Russia; olbazanova@iopc.ru (O.B.); babaev@iopc.ru (V.B.); aplyubina@gmail.com (A.L.); sobaka-1968@mail.ru (A.V.); kpetrov2005@mail.ru (K.P.)

**Keywords:** thiacalixarene, synthesis, antibacterial activity, Gram-positive bacteria, cytotoxicity, antibacterial agent

## Abstract

Antimicrobial resistance to modern antibiotics stimulates the search for new ways to synthesize and modify antimicrobial drugs. The development of synthetic approaches that can easily change different fragments of the molecule is a promising solution to this problem. In this work, a synthetic approach was developed to obtain multivalent thiacalix[4]arene derivatives containing different number of amine and hydroxyl groups. A series of macrocyclic compounds in *cone*, *partial cone*, and *1*,*3-alternate* stereoisomeric forms containing -NHCH_2_CH_2_R (R = NH_2_, N(CH_3_)_2_, and OH) and -N(CH_2_CH_2_OH)_2_ terminal fragments, and their model non-macrocyclic analogues were obtained. The antibacterial activity against Gram-positive (*Staphylococcus aureus*, *Bacillus cereus*, and *Enterococcus faecalis*) and Gram-negative (*Escherichia coli* and *Pseudomonas aeruginosa*) bacterial strains and cytotoxicity of the obtained compounds were studied. Structure–activity relationships were established: (1) the macrocyclic compounds had high antibacterial activity, while the monomeric compounds had low activity; (2) the compounds in *cone* and *partial cone* conformations had better antibacterial activity compared to the compounds in *1*,*3-alternate* stereoisomeric form; (3) the macrocyclic compounds containing -NHCH_2_CH_2_N(CH_3_)_2_ terminal fragments had the highest antibacterial activity; (4) introduction of additional terminal hydroxyl groups led to a significant decrease in antibacterial activity; (5) the compounds in *partial cone* conformation had significant bactericidal activity against all studied cell strains; the best selectivity was observed for the compounds in *cone* conformation. The mechanism of antibacterial activity of lead compounds with terminal fragments -NHCH_2_CH_2_N(CH_3_)_2_ was proved using model negatively charged POPG vesicles, i.e., the addition of these compounds led to an increase in the size and zeta potential of the vesicles. The obtained results open up the possibility of using the synthesized macrocyclic compounds as promising antibacterial agents.

## 1. Introduction

According to the World Health Organization [1], antimicrobial resistance to modern antibiotics is one of the most important problems of the 21st century. Bacteria have several mechanisms to combat antibiotics, such as the formation of biofilms, membrane changes, the production of enzymes capable of modifying antibiotics, and chromosomal mutations [2,3,4,5]. The presence of such a large number of mechanisms leads to difficulties in the synthesis of new effective antibacterial agents [6]. One of the possible solutions to this problem is the use of multivalent compounds, which, due to different functional groups, are able to interact effectively with the bacterial cell membrane, leading to its destabilization and destruction [7,8,9,10,11,12,13,14]. Linear [15,16,17,18] and hyperbranched polymeric macromolecules [19,20,21,22,23,24,25,26,27,28,29] with low symmetry, as well as more symmetric dendrimers with a large number of positively charged fragments [30,31,32,33,34,35,36,37,38,39,40,41], are often used as such multivalent structures.

A problem in the design and synthesis of hyperbranched structures is usually the difficulty in controlling their hydrophilic–lipophilic balance, which is an important factor in “tuning” the macromolecule. Introduction of a lipophilic macrocyclic core, such as (thia)calix[n]arenes, into the structure of the target compounds leads to changes in the physical and biological properties of the resulting compounds [42,43,44,45,46,47,48,49,50,51]. There are a number of examples in the literature of antimicrobial agents based on these macrocyclic compounds [52,53,54,55,56,57,58]. A recent review by Prof. Duval’s group [59] detailed the benefits of the (thia)calixarene platforms, e.g., the use of a multivalent approach to obtain the target compounds, the possibility of easy functionalization with positively charged fragments (e.g., ammonium, guanidinium, and imidazolium), and the spatial organization of these fragments by obtaining different stereoisomeric forms, i.e., *cone*, *partial cone*, and *1*,*3-altenate*. Previously, our research group has shown that thiacalix[4]arene derivatives containing eight primary terminal amino groups and their complexes with lysozyme have antibacterial properties [60]. However, the synthetic approaches proposed earlier in the literature do not allow for the possibility of “fine-tuning” the structure of the target compounds by the introduction of different functional groups.

In this work, a synthetic approach to obtaining multivalent thiacalix[4]arene derivatives containing different amounts of amine and hydroxyl groups at the lower rim was proposed and developed. The antibacterial properties of the synthesized compounds were also evaluated, and structure–activity relationships were established. The obtained results open up the possibility of using these compounds as antibacterial agents.

## 2. Materials and Methods

### 2.1. General Experimental Information

More details on the equipment, methods of confirmation, and establishment of the compounds structure (NMR, IR, and mass spectra of the synthesized compounds, antimicrobial and cytotoxic assay, dynamic light scattering data) are described in the Appendix A.

Most chemicals (acryloyl chloride, triethylamine, ethylenediamine, *N*,*N*-dimethylethylenediamine, ethanolamine, and diethanolamine) were purchased from Sigma-Aldrich, Dortmund, Germany. TRIS buffer (pH = 7.4, 150 mM NaCl) was purchased from Fisher Scientific. Organic solvents were purified in accordance with standard procedures. Deionized water with resistivity >18.0 MΩ cm (Millipore-Q, Simplicity® water purification system, Merck-Millipore, Molsheim, France) was used for the solutions preparation. 1-Palmitoyl-2-oleoyl-*sn*-glycero-3-phospho-(1′-*rac*-glycerol) sodium salt (POPG) was purchased from Avanti Polar Lipids, Birmingham, AL, USA.

Thiacalix[4]arene derivatives **1**–**3** (*cone*, *partial cone*, and *1*,*3-alternate* conformations, respectively) and monomer analogue (*p-tert*-butylphenol derivative) **19** were synthesized by the previously described procedure [61].

### 2.2. General Procedure for the Synthesis of Compounds ***4***–***6***

The solution of acryloyl chloride (0.27 mL, 3.27 mmol) in 10 mL of corresponding solvent was added dropwise to an ice-cooled (0 °C) mixture of compounds **1**–**3** (1.00 g, 0.74 mmol) and triethylamine (0.93 mL, 6.89 mmol) in 20 mL of CH_2_Cl_2_ (for compounds **1** or **3**) or CHCl_3_ (for compound **2**). The reaction mixture was stirred for 12 h at room temperature. Afterward, the solvent was removed on a rotary evaporator, and the residue was washed with water (3 × 20 mL). Then, the wet residue was dissolved in a minimal amount of hot ethanol and poured into 25 mL of water. The resulting suspension was centrifuged, and the precipitate was dried under reduced pressure.

#### 2.2.1. 5,11,17,23-Tetra-*tert*-butyl-25,26,27,28-tetrakis[*N*-(6-(acrylamido)hexyl)carbamoylmethoxy]-2,8,14,20-tetrathiacalix[4]arene **4** in *cone* conformation. Yield: 0.98 g (85%). White Powder, mp 88 °C

^1^H NMR (DMSO-*d*_6_, δ, ppm, *J*/Hz): 1.06 (s, 36H, (CH_3_)_3_C), 1.24 (m, 16H, C(O)NHCH_2_CH_2_CH_2_CH_2_), 1.40 (m, 8H, CH_2_CH_2_CH_2_NH), 1.45 (m, 8H, C(O)NHCH_2_CH_2_CH_2_), 3.05–3.13 (m, 8H, CH_2_NHC(O)CH=CH_2_), 3.13–3.19 (m, 8H, OCH_2_C(O)NHCH_2_), 4.76 (s, 8H, OCH_2_C(O)), 5.54 (dd, 4H, part of ABX system, CH=CH_2_, ^2^*J*_HH_ = 2.5, ^3^*J*_HH_ = 10.0), 6.05 (dd, 4H, part of ABX system, CH=CH_2_, ^2^*J*_HH_ = 2.5, ^3^*J*_HH_ = 17.1), 6.19 (dd, 4H, part of ABX system, CH=CH_2_, ^3^*J*_HH_ = 10.0, ^3^*J*_HH_ = 17.1), 7.38 (s, 8H, ArH), 8.05 (br.t, 4H, NHC(O)CH=CH_2_), 8.34 (br.t, 4H, OCH_2_CONH).

^13^C{^1^H} NMR (DMSO-*d*_6_, δ, ppm): 26.22, 26.25, 29.05, 29.11, 30.73, 33.89, 38.43, 38.49, 73.81, 124.73, 128.00, 131.88, 134.37, 146.48, 157.70, 164.41, 167.67.

FTIR ATR (ν, cm^−1^): 3287 (N-H), 3075 (N-H), 1653 (C(O)NH, amide I), 1624 (C=C), 1540 (C(O)NH, amide II), 1093 (C_Ph_OCH_2_).

ESI-HRMS, Calculated [M + H + NEt_3_]^+^ *m*/*z* = 1663.9219, [M + Na]^+^ *m*/*z* = 1583.7801, [M + 2H + NEt_3_]^2+^ *m*/*z* = 831.9629, [M + 2H]^2+^ *m*/*z* = 781.4027. Found [M + H + NEt_3_]^+^ *m*/*z* = 1663.9155, [M + Na]^+^ *m*/*z* = 1583.7760, [M + 2H + NEt_3_]^2+^ *m*/*z* = 831.9606, [M + 2H]^2+^ *m*/*z* = 781.3992.

#### 2.2.2. 5,11,17,23-Tetra-*tert*-butyl-25,26,27,28-tetrakis[*N*-(6-(acrylamido)hexyl)carbamoylmethoxy]-2,8,14,20-tetrathiacalix[4]arene **5** in *partial cone* conformation. Yield: 1.11 g (96%). White Powder, mp 84 °C

^1^H NMR (DMSO-*d*_6_, δ, ppm, *J*/Hz): 1.00 (s, 18H, (CH_3_)_3_C), 1.12–1.54 (m, 32H, C(O)NHCH_2_CH_2_CH_2_CH_2_, CH_2_CH_2_CH_2_NH, C(O)NHCH_2_CH_2_CH_2_), 1.25 (s, 9H, (CH_3_)_3_C), 1.27 (s, 9H, (CH_3_)_3_C), 2.95–3.25 (m, 16H, CH_2_NHC(O)CH=CH_2_, OCH_2_C(O)NHCH_2_), 4.10 (d, 2H, OCH_2_C(O), ^2^*J*_HH_ = 13.5), 4.47 (s, 2H, OCH_2_C(O)), 4.57 (s, 2H, OCH_2_C(O)), 4.87 (d, 2H, OCH_2_C(O), ^2^*J*_HH_ = 13.5), 5.54 (dd, 4H, part of ABX system, CH=CH_2_, ^2^*J*_HH_ = 2.3, ^3^*J*_HH_ = 10.0), 6.04 (dd, 4H, part of ABX system, CH=CH_2_, ^2^*J*_HH_ = 2.3, ^3^*J*_HH_ = 17.1), 6.13–6.24 (m, 4H, part of ABX system), 6.99 (d, 2H, ArH, ^2^*J*_HH_ = 2.5), 7.58 (s, 2H, ArH), 7.62 (d, 2H, ArH, ^2^*J*_HH_ = 2.5), 7.69 (s, 2H, ArH), 8.05 (br.t, 4H, NHC(O)CH=CH_2_), 8.18–8.24 (m, 3H, OCH_2_CONH), 8.18–8.24 (br.t, 1H, OCH_2_CONH).

^13^C{^1^H} NMR (DMSO-*d*_6_, δ, ppm): 26.15, 26.26, 28.99, 29.08, 29.14, 30.75, 31.04, 33.81, 33.85, 33.90, 38.18, 38.51, 38.56, 68.90, 72.66, 72.89, 124.79, 126.45, 126.87, 127.38, 128.01, 131.90, 133.62, 133.96, 134.30, 135.49, 144.60, 145.44, 146.61, 156.52, 157.59, 159.45, 164.47, 166.79, 167.44, 168.08.

FTIR ATR (ν, cm^−1^): 3285 (N-H), 3074 (N-H), 1652 (C(O)NH, amide I), 1625 (C=C), 1538 (C(O)NH, amide II), 1088 (C_Ph_OCH_2_).

ESI-HRMS, Calculated [M + H]^+^ *m*/*z* = 1562.8015, [M + Na]^+^ *m*/*z* = 1583.7801, [M + 2H]^2+^ *m*/*z* = 781.4027. Found [M + H]^+^ *m*/*z* = 1562.7946, [M + Na]^+^ *m*/*z* = 1583.7714, [M + 2H]^2+^ *m*/*z* = 781.4010.

#### 2.2.3. 5,11,17,23-Tetra-*tert*-butyl-25,26,27,28-tetrakis[*N*-(6-(acrylamido)hexyl)carbamoylmethoxy]-2,8,14,20-tetrathiacalix[4]arene **6** in *1*,*3-alternate* conformation. Yield: 1.07 g (92%). White Powder, mp 91 °C

^1^H NMR (DMSO-*d*_6_, δ, ppm, *J*/Hz): 1.18 (s, 36H, (CH_3_)_3_C), 1.25 (m, 16H, C(O)NHCH_2_CH_2_CH_2_CH_2_), 1.36–1.46 (m, 16H, CH_2_CH_2_CH_2_NH, C(O)NHCH_2_CH_2_CH_2_), 3.02–3.14 (m, 16H, CH_2_NHC(O)CH=CH_2_, OCH_2_C(O)NHCH_2_), 3.93 (s, 8H, OCH_2_C(O)), 5.55 (dd, 4H, part of ABX system, CH=CH_2_, ^2^*J*_HH_ = 2.5, ^3^*J*_HH_ = 10.0), 6.05 (dd, 4H, part of ABX system, CH=CH_2_, ^2^*J*_HH_ = 2.5, ^3^*J*_HH_ = 17.1), 6.19 (dd, 4H, part of ABX system, CH=CH_2_, ^3^*J*_HH_ = 10.0, ^3^*J*_HH_ = 17.1), 7.51 (s, 8H, ArH), 7.69 (br.t, 4H, OCH_2_CONH), 8.05 (br.t, 4H, NHC(O)CH=CH_2_).

^13^C{^1^H} NMR (DMSO-*d*_6_, δ, ppm): 26.20, 26.30, 29.04, 29.10, 30.74, 33.86, 38.45, 38.72, 70.40, 124.75, 127.47, 131.89, 132.07, 146.23, 156.52, 164.41, 166.87.

FTIR ATR (ν, cm^−1^): 3297 (N-H), 3072 (N-H), 1653 (C(O)NH, amide I), 1626 (C=C), 1533 (C(O)NH, amide II), 1086 (C_Ph_OCH_2_).

ESI-HRMS, Calculated [M + H]^+^ *m*/*z* = 1562.8015, [M + 2H]^2+^ *m*/*z* = 781.4027. Found [M + H]^+^ *m*/*z* = 1562.7938, [M + 2H]^2+^ *m*/*z* = 781.4122.

### 2.3. General Procedure for the Synthesis of Compounds ***7***–***12***

The corresponding diamine (1.92 mmol) (ethylenediamine for **7**–**9** and *N*,*N*-dimethylethylenediamine for **10**–**12**) was added to the solution of **4**–**6** (0.10 g, 0.064 mmol) in 7 mL of methanol. The reaction mixture was refluxed for 60 h (for **7**–**9**) or 90 h (for **10**–**12**). Then the solvent was evaporated under reduced pressure, and the remaining diamine was removed by azeotropic distillation (toluene:methanol mixture, 9:1). Afterward, the remaining toluene was removed by azeotropic distillation with methanol. The residue was dried under reduced pressure.

#### 2.3.1. 5,11,17,23-Tetra-*tert*-butyl-25,26,27,28-tetrakis[*N*-(6-(*N*-(3-(2-aminoethyl)aminopropanoyl)amino)hexyl)carbamoylmethoxy]-2,8,14,20-tetrathiacalix[4]arene **7** in *cone* conformation. Yield: 0.11 g (97%). White Solid Foam, mp 90 °C

^1^H NMR (DMSO-*d*_6_, δ, ppm, *J*/Hz): 1.07 (s, 36H, (CH_3_)_3_C), 1.23 (m, 16H, C(O)NHCH_2_CH_2_CH_2_CH_2_), 1.36 (m, 8H, CH_2_CH_2_CH_2_NH), 1.45 (m, 8H, C(O)NHCH_2_CH_2_CH_2_), 2.18 (t, 8H, NHCH_2_CH_2_C(O), ^3^*J*_HH_ = 6.8), 2.47 (m, 8H, NH_2_CH_2_CH_2_NH), 2.57 (m, 8H, NH_2_CH_2_CH_2_NH), 2.66 (t, 8H, NHCH_2_CH_2_C(O), ^3^*J*_HH_ = 6.8), 2.96–3.05 (m, 8H, CH_2_NHC(O)CH_2_CH_2_), 3.10–3.22 (m, 8H, OCH_2_C(O)NHCH_2_), 4.75 (s, 8H, OCH_2_C(O)), 7.38 (s, 8H, ArH), 7.92 (br.t, 4H, NHC(O)CH_2_CH_2_), 8.34 (br.t, 4H, OCH_2_CONH).

^13^C{^1^H} NMR (DMSO-*d*_6_, δ, ppm): 26.19, 29.10, 30.70, 33.84, 35.95, 38.28, 38.42, 41.13, 45.51, 51.69, 73.79, 127.98, 134.34, 146.45, 157.68, 167.62, 171.15.

FTIR ATR (ν, cm^−1^): 3287 (N-H), 1649 (C(O)NH, amide I), 1545 (C(O)NH, amide II), 1094 (C_Ph_OCH_2_).

ESI-HRMS, Calculated [M + Na]^+^ *m*/*z* = 1825.0584, [M + 2H]^2+^ *m*/*z* = 902.0419, [M + 3H]^3+^ *m*/*z* = 601.6970. Found [M + Na]^+^ *m*/*z* = 1825.0580, [M + 2H]^2+^ *m*/*z* = 902.0439, [M + 3H]^3+^ *m*/*z* = 601.6988.

#### 2.3.2. 5,11,17,23-Tetra-*tert*-butyl-25,26,27,28-tetrakis[*N*-(6-(*N*-(3-(2-aminoethyl)aminopropanoyl)amino)hexyl)carbamoylmethoxy]-2,8,14,20-tetrathiacalix[4]arene **8** in *partial cone* conformation. Yield: 0.11 g (92%). White Solid Foam, mp 88 °C

^1^H NMR (DMSO-*d*_6_, δ, ppm, *J*/Hz): 1.00 (s, 18H, (CH_3_)_3_C), 1.13–1.53 (m, 32H, C(O)NHCH_2_CH_2_CH_2_CH_2_, CH_2_CH_2_CH_2_NH, C(O)NHCH_2_CH_2_CH_2_), 1.26 (s, 9H, (CH_3_)_3_C), 1.29 (s, 9H, (CH_3_)_3_C), 2.18 (br.t, 8H, NHCH_2_CH_2_CO), 2.48 (m, 8H, NH_2_CH_2_CH_2_NH), 2.58 (m, 8H, NH_2_CH_2_CH_2_NH), 2.66 (br.t, 8H, NHCH_2_CH_2_C(O)), 2.94–3.13 (m, 16H, CH_2_NHC(O)CH_2_CH_2_, OCH_2_C(O)NHCH_2_), 4.11 (d, 2H, OCH_2_C(O), ^2^*J*_HH_ = 13.4), 4.47 (s, 2H, OCH_2_C(O)), 4.58 (s, 2H, OCH_2_C(O)), 4.86 (d, 2H, OCH_2_C(O), ^2^*J*_HH_ = 13.4), 7.00 (br.d, 2H, ArH), 7.59 (s, 2H, ArH), 7.62 (br.d, 2H, ArH), 7.70 (s, 2H, ArH), 7.92 (br.t, 4H, NHC(O)CH_2_CH_2_), 8.21 (m, 3H, OCH_2_CONH), 8.18–8.24 (br.t, 1H, OCH_2_CONH).

^13^C{^1^H} NMR (DMSO-*d*_6_, δ, ppm): 26.20, 26.26, 28.96, 29.06, 29.12, 30.72, 30.77, 31.01, 33.76, 33.86, 35.74, 38.18, 38.33, 38.53, 45.24, 49.40, 68.92, 72.64, 72.84, 126.39, 126.82, 127.38, 127.96, 133.57, 133.91, 134.28, 135.46, 144.60, 145.36, 146.56, 156.50, 157.56, 159.41, 166.74, 167.38, 167.99, 171.11.

FTIR ATR (ν, cm^−1^): 3279 (N-H), 3064 (N-H), 1646 (C(O)NH, amide I), 1545 (C(O)NH, amide II), 1089 (C_Ph_OCH_2_).

ESI-HRMS, Calculated [M + 2H]^2+^ *m*/*z* = 902.0419, [M + 3H]^3+^ *m*/*z* = 601.6970. Found [M + 2H]^2+^ *m*/*z* = 902.0443, [M + 3H]^3+^ *m*/*z* = 601.6992.

#### 2.3.3. 5,11,17,23-Tetra-*tert*-butyl-25,26,27,28-tetrakis[*N*-(6-(*N*-(3-(2-aminoethyl)aminopropanoyl)amino)hexyl)carbamoylmethoxy]-2,8,14,20-tetrathiacalix[4]arene **9** in *1*,*3-alternate* conformation. Yield: 0.10 g (90%). White Solid Foam, mp 88 °C

^1^H NMR (DMSO-*d*_6_, δ, ppm, *J*/Hz): 1.19 (s, 36H, (CH_3_)_3_C), 1.24 (m, 16H, C(O)NHCH_2_CH_2_CH_2_CH_2_), 1.37 (m, 8H, CH_2_CH_2_CH_2_NH), 1.43 (m, 8H, C(O)NHCH_2_CH_2_CH_2_), 2.18 (t, 8H, NHCH_2_CH_2_C(O), ^3^*J*_HH_ = 6.8), 2.47 (m, 8H, NH_2_CH_2_CH_2_NH), 2.56 (m, 8H, NH_2_CH_2_CH_2_NH), 2.66 (t, 8H, NHCH_2_CH_2_C(O), ^3^*J*_HH_ = 6.8), 2.96–3.03 (m, 8H, CH_2_NHC(O)CH_2_CH_2_), 3.04–3.11 (m, 8H, OCH_2_C(O)NHCH_2_), 3.93 (s, 8H, OCH_2_C(O)), 7.52 (s, 8H, ArH), 7.70 (br.t, 4H, OCH_2_CONH), 7.92 (br.t, 4H, NHC(O)CH_2_CH_2_).

^13^C{^1^H} NMR (DMSO-*d*_6_, δ, ppm): 26.18, 26.30, 29.13, 30.75, 33.86, 36.04, 38.27, 38.72, 41.31, 45.62, 52.01, 70.45, 127.47, 132.11, 146.22, 156.54, 166.87, 171.17.

FTIR ATR (ν, cm^−1^): 3299 (N-H), 3064 (N-H), 1646 (C(O)NH, amide I), 1534 (C(O)NH, amide II), 1085 (C_Ph_OCH_2_).

ESI-HRMS, Calculated [M + Na]^+^ *m*/*z* = 1825.0584, [M + 2H]^2+^ *m*/*z* = 902.0419, [M + 3H]^3+^ *m*/*z* = 601.6970, [M + 4H]^4+^ *m*/*z* = 451.5246. Found [M + Na]^+^ *m*/*z* = 1825.0567, [M + 2H]^2+^ *m*/*z* = 902.0445, [M + 3H]^3+^ *m*/*z* = 601.6989, [M + 4H]^4+^ *m*/*z* = 451.5266.

#### 2.3.4. 5,11,17,23-Tetra-*tert*-butyl-25,26,27,28-tetrakis[*N*-(6-(*N*-(3-(2-(*N*,*N*-dimethylamino)ethyl)aminopropanoyl)amino)hexyl)carbamoylmethoxy]-2,8,14,20-tetrathiacalix[4]arene **10** in *cone* conformation. Yield: 0.12 g (98%). White Solid Foam, mp 60 °C

^1^H NMR (DMSO-*d*_6_, δ, ppm, *J*/Hz): 1.06 (s, 36H, (CH_3_)_3_C), 1.23 (m, 16H, C(O)NHCH_2_CH_2_CH_2_CH_2_), 1.35 (m, 8H, CH_2_CH_2_CH_2_NH), 1.45 (m, 8H, C(O)NHCH_2_CH_2_CH_2_), 2.08 (s, 24H, N(CH_3_)_2_), 2.18 (t, 8H, NHCH_2_CH_2_C(O), ^3^*J*_HH_ = 6.7), 2.25 (br.t, 8H, (CH_3_)_2_NCH_2_CH_2_NH), 2.53 (br.t, 8H, (CH_3_)_2_NCH_2_CH_2_NH), 2.66 (t, 8H, NHCH_2_CH_2_C(O), ^3^*J*_HH_ = 6.7), 2.95–3.04 (m, 8H, CH_2_NHC(O)CH_2_CH_2_), 3.10–3.20 (m, 8H, OCH_2_C(O)NHCH_2_), 4.75 (s, 8H, OCH_2_C(O)), 7.37 (s, 8H, ArH), 7.94 (br.t, 4H, NHC(O)CH_2_CH_2_), 8.34 (br.t, 4H, OCH_2_CONH).

^13^C{^1^H} NMR (DMSO-*d*_6_, δ, ppm): 26.26, 29.17, 30.75, 33.90, 35.80, 38.31, 38.45, 45.27, 45.75, 46.65, 58.79, 73.84, 128.02, 134.38, 146.48, 157.74, 167.66, 171.14.

FTIR ATR (ν, cm^−1^): 3300 (N-H), 3073 (N-H), 1648 (C(O)NH, amide I), 1542 (C(O)NH, amide II), 1095 (C_Ph_OCH_2_).

ESI-HRMS, Calculated [M + Na]^+^ *m*/*z* = 1937.1836, [M + 2H]^2+^ *m*/*z* = 958.1045, [M + 3H]^3+^ *m*/*z* = 639.0721. Found [M + Na]^+^ *m*/*z* = 1937.1793, [M + 2H]^2+^ *m*/*z* = 958.1057, [M + 3H]^3+^ *m*/*z* = 639.0736.

#### 2.3.5. 5,11,17,23-Tetra-*tert*-butyl-25,26,27,28-tetrakis[*N*-(6-(*N*-(3-(2-(*N*,*N*-dimethylamino)ethyl)aminopropanoyl)amino)hexyl)carbamoylmethoxy]-2,8,14,20-tetrathiacalix[4]arene **11** in *partial cone* conformation. Yield: 0.11 g (94%). White Solid Foam, mp 66 °C

^1^H NMR (DMSO-*d*_6_, δ, ppm, *J*/Hz): 1.00 (s, 18H, (CH_3_)_3_C), 1.13–1.54 (m, 32H, C(O)NHCH_2_CH_2_CH_2_CH_2_, CH_2_CH_2_CH_2_NH, C(O)NHCH_2_CH_2_CH_2_), 1.26 (s, 9H, (CH_3_)_3_C), 1.28 (s, 9H, (CH_3_)_3_C), 2.09 (s, 24H, N(CH_3_)_2_), 2.19 (br.t, 8H, NHCH_2_CH_2_C(O)), 2.27 (br.t, 8H, (CH_3_)_2_NCH_2_CH_2_NH), 2.55 (br.t, 8H, (CH_3_)_2_NCH_2_CH_2_NH), 2.70 (br.t, 8H, NHCH_2_CH_2_C(O)), 2.96–3.22 (m, 16H, CH_2_NHC(O)CH_2_CH_2_, OCH_2_C(O)NHCH_2_), 4.11 (d, 2H, OCH_2_C(O), ^2^*J*_HH_ = 13.4), 4.46 (s, 2H, OCH_2_C(O)), 4.58 (s, 2H, OCH_2_C(O)), 4.86 (d, 2H, OCH_2_C(O), ^2^*J*_HH_ = 13.4), 6.99 (br.d, 2H, ArH), 7.59 (s, 2H, ArH), 7.62 (br.d, 2H, ArH), 7.70 (s, 2H, ArH), 7.93 (br.t, 4H, NHC(O)CH_2_CH_2_), 8.20 (m, 3H, OCH_2_CONH), 8.29 (br.t, 1H, OCH_2_CONH).

^13^C{^1^H} NMR (DMSO-*d*_6_, δ, ppm): 26.21, 26.29, 28.98, 29.14, 29.22, 30.74, 30.79, 31.03, 33.78, 35.17, 38.17, 38.31, 38.38, 38.54, 45.20, 45.43, 46.29, 58.21, 68.91, 72.64, 72.85, 126.42, 126.84, 127.37, 127.97, 133.60, 133.92, 134.28, 135.48, 144.56, 145.36, 146.56, 156.51, 157.57, 159.44, 166.73, 167.37, 168.00, 170.90.

FTIR ATR (ν, cm^−1^): 3290 (N-H), 3072 (N-H), 1648 (C(O)NH, amide I), 1542 (C(O)NH, amide II), 1091 (C_Ph_OCH_2_).

ESI-HRMS, Calculated [M + Na]^+^ *m*/*z* = 1937.1836, [M + 2H]^2+^ *m*/*z* = 958.1045, [M + 3H]^3+^ *m*/*z* = 639.0721, [M + 4H]^4+^ *m*/*z* = 479.5559. Found [M + Na]^+^ *m*/*z* = 1937.1832, [M + 2H]^2+^ *m*/*z* = 958.1070, [M + 3H]^3+^ *m*/*z* = 639.0745, [M + 4H]^4+^ *m*/*z* = 479.5578.

#### 2.3.6. 5,11,17,23-Tetra-*tert*-butyl-25,26,27,28-tetrakis[*N*-(6-(*N*-(3-(2-(*N*,*N*-dimethylamino)ethyl)aminopropanoyl)amino)hexyl)carbamoylmethoxy]-2,8,14,20-tetrathiacalix[4]arene **12** in *1*,*3-alternate* conformation. Yield: 0.12 g (97%). White Solid Foam, mp 62 °C

^1^H NMR (DMSO-*d*_6_, δ, ppm, *J*/Hz): 1.19 (s, 36H, (CH_3_)_3_C), 1.20–1.26 (m, 16H, C(O)NHCH_2_CH_2_CH_2_CH_2_), 1.36 (m, 8H, CH_2_CH_2_CH_2_NH), 1.43 (m, 8H, C(O)NHCH_2_CH_2_CH_2_), 2.09 (s, 24H, N(CH_3_)_2_), 2.18 (br.t, 8H, NHCH_2_CH_2_C(O)), 2.25 (br.t, 8H, (CH_3_)_2_NCH_2_CH_2_NH), 2.54 (br.t, 8H, (CH_3_)_2_NCH_2_CH_2_NH), 2.69 (br.t, 8H, NHCH_2_CH_2_C(O)), 2.96–3.02 (m, 8H, CH_2_NHC(O)CH_2_CH_2_), 3.03–3.12 (m, 8H, OCH_2_C(O)NHCH_2_), 3.92 (s, 8H, OCH_2_C(O)), 7.51 (s, 8H, ArH), 7.70 (br.t, 4H, OCH_2_CONH), 7.95 (br.t, 4H, NHC(O)CH_2_CH_2_).

^13^C{^1^H} NMR (DMSO-*d*_6_, δ, ppm): 26.20, 26.34, 29.15, 30.77, 33.87, 35.82, 38.28, 38.74, 45.29, 45.76, 46.65, 58.80, 70.42, 127.49, 132.11, 146.23, 156.54, 166.87, 171.13.

FTIR ATR (ν, cm^−1^): 3299 (N-H), 3072 (N-H), 1647 (C(O)NH, amide I), 1536 (C(O)NH, amide II), 1086 (C_Ph_OCH_2_).

ESI-HRMS, Calculated [M + Na]^+^ *m*/*z* = 1937.1836, [M + 2H]^2+^ *m*/*z* = 958.1045, [M + 3H]^3+^ *m*/*z* = 639.0721, [M + 4H]^4+^ *m*/*z* = 479.5559. Found [M + Na]^+^ *m*/*z* = 1937.1816, [M + 2H]^2+^ *m*/*z* = 958.1060, [M + 3H]^3+^ *m*/*z* = 639.0745, [M + 4H]^4+^ *m*/*z* = 479.5575.

### 2.4. General Procedure for the Synthesis of Compounds ***13***–***15***

Ethanolamine (0.23 mL, 3.84 mmol) was added to the solution of **4**–**6** (0.20 g, 0.128 mmol) in methanol (10 mL). The reaction mixture was refluxed for 90 h. Afterward, the solvent was evaporated under reduced pressure. Then 15 mL of water was added to the residue, and the resulting suspension was centrifuged. The precipitate was dried under reduced pressure.

#### 2.4.1. 5,11,17,23-Tetra-*tert*-butyl-25,26,27,28-tetrakis[*N*-(6-(*N*-(3-(2-hydroxyethyl)aminopropanoyl)amino)hexyl)carbamoylmethoxy]-2,8,14,20-tetrathiacalix[4]arene **13** in *cone* conformation. Yield: 0.19 g (84%). White Powder, mp 72 °C

^1^H NMR (DMSO-*d*_6_, δ, ppm, *J*/Hz): 1.07 (s, 36H, (CH_3_)_3_C), 1.23 (m, 16H, C(O)NHCH_2_CH_2_CH_2_CH_2_), 1.36 (m, 8H, CH_2_CH_2_CH_2_NH), 1.44 (m, 8H, C(O)NHCH_2_CH_2_CH_2_), 2.18 (t, 8H, NHCH_2_CH_2_C(O), ^3^*J*_HH_ = 6.8), 2.53 (t, 8H, NHCH_2_CH_2_OH, ^3^*J*_HH_ = 5.8), 2.68 (t, 8H, NHCH_2_CH_2_C(O), ^3^*J*_HH_ = 6.8), 2.95–3.03 (m, 8H, CH_2_NHC(O)CH_2_CH_2_), 3.11–3.19 (m, 8H, OCH_2_C(O)NHCH_2_), 3.41 (t, 8H, NHCH_2_CH_2_OH, ^3^*J*_HH_ = 5.8), 4.75 (s, 8H, OCH_2_C(O)), 7.38 (s, 8H, ArH), 7.90 (br.t, 4H, NHC(O)CH_2_CH_2_), 8.33 (br.t, 4H, OCH_2_CONH).

^13^C{^1^H} NMR (DMSO-*d*_6_, δ, ppm): 26.25, 29.15, 30.74, 33.90, 35.91, 38.33, 38.45, 45.67, 51.47, 60.30, 73.82, 128.02, 134.38, 146.48, 157.73, 167.67, 171.14.

FTIR ATR (ν, cm^−1^): 3299 (N-H), 3076 (N-H), 1646 (C(O)NH, amide I), 1542 (C(O)NH, amide II), 1094 (C_Ph_OCH_2_).

ESI-HRMS, Calculated [M + 2H]^2+^ *m*/*z* = 904.0099, [M + 3H]^3+^ *m*/*z* = 603.0090. Found [M + 2H]^2+^ *m*/*z* = 904.0121, [M + 3H]^3+^ *m*/*z* = 603.0109.

#### 2.4.2. 5,11,17,23-Tetra-*tert*-butyl-25,26,27,28-tetrakis[*N*-(6-(*N*-(3-(2-hydroxyethyl)aminopropanoyl)amino)hexyl)carbamoylmethoxy]-2,8,14,20-tetrathiacalix[4]arene **14** in *partial cone* conformation. Yield: 0.19 g (82%). White Powder, mp 65 °C

^1^H NMR (DMSO-*d*_6_, δ, ppm, *J*/Hz): 1.00 (s, 18H, (CH_3_)_3_C), 1.12–1.53 (m, 32H, C(O)NHCH_2_CH_2_CH_2_CH_2_, CH_2_CH_2_CH_2_NH, C(O)NHCH_2_CH_2_CH_2_), 1.26 (s, 9H, (CH_3_)_3_C), 1.28 (s, 9H, (CH_3_)_3_C), 2.19 (br.t, 8H, NHCH_2_CH_2_C(O)), 2.54 (br.t, 8H, NHCH_2_CH_2_OH), 2.68 (br.t, 8H, NHCH_2_CH_2_C(O)), 2.94–3.03 (m, 8H, CH_2_NHC(O)CH_2_CH_2_), 3.04–3.12 (m, 8H, OCH_2_C(O)NHCH_2_), 3.41 (br.t, 8H, NHCH_2_CH_2_OH), 2.95–3.03 (m, 8H, CH_2_NHC(O)CH_2_CH_2_), 3.05–3.23 (m, 8H, OCH_2_C(O)NHCH_2_), 4.11 (d, 2H, OCH_2_C(O), ^2^*J*_HH_ = 13.4), 4.47 (s, 2H, OCH_2_C(O)), 4.57 (s, 2H, OCH_2_C(O)), 4.86 (d, 2H, OCH_2_C(O), ^2^*J*_HH_ = 13.4), 6.99 (br.d, 2H, ArH), 7.59 (s, 2H, ArH), 7.62 (br.d, 2H, ArH), 7.70 (s, 2H, ArH), 7.90 (br.t, 4H, NHC(O)CH_2_CH_2_), 8.20 (m, 3H, OCH_2_CONH), 8.29 (br.t, 1H, OCH_2_CONH).

^13^C{^1^H} NMR (DMSO-*d*_6_, δ, ppm): 26.22, 26.27, 28.97, 29.07, 29.14, 30.73, 30.78, 31.02, 33.78, 33.82, 33.89, 35.92, 38.18, 38.32, 38.37, 38.55, 45.68, 51.47, 60.30, 68.94, 72.66, 72.87, 126.40, 126.84, 127.38, 127.98, 133.59, 133.93, 134.30, 135.48, 144.61, 145.39, 146.57, 156.51, 157.57, 159.43, 166.75, 167.39, 168.00, 171.14.

FTIR ATR (ν, cm^−1^): 3290 (N-H), 3076 (N-H), 1648 (C(O)NH, amide I), 1540 (C(O)NH, amide II), 1088 (C_Ph_OCH_2_).

ESI-HRMS, Calculated [M + Na]^+^ *m*/*z* = 1828.9945, [M + 2H]^2+^ *m*/*z* = 904.0099, [M + 3H]^3+^ *m*/*z* = 603.0090. Found [M + Na]^+^ *m*/*z* = 1828.9949, [M + 2H]^2+^ *m*/*z* = 904.0124, [M + 3H]^3+^ *m*/*z* = 603.0111.

#### 2.4.3. 5,11,17,23-Tetra-*tert*-butyl-25,26,27,28-tetrakis[*N*-(6-(*N*-(3-(2-hydroxyethyl)aminopropanoyl)amino)hexyl)carbamoylmethoxy]-2,8,14,20-tetrathiacalix[4]arene **15** in *1*,*3-alternate* conformation. Yield: 0.18 g (79%). White Powder, mp 80 °C

^1^H NMR (DMSO-*d*_6_, δ, ppm, *J*/Hz): 1.19 (s, 36H, (CH_3_)_3_C), 1.22 (m, 16H, C(O)NHCH_2_CH_2_CH_2_CH_2_), 1.37 (m, 8H, CH_2_CH_2_CH_2_NH), 1.43 (m, 8H, C(O)NHCH_2_CH_2_CH_2_), 2.19 (t, 8H, NHCH_2_CH_2_C(O), ^3^*J*_HH_ = 6.8), 2.54 (t, 8H, NHCH_2_CH_2_OH, ^3^*J*_HH_ = 5.7), 2.68 (t, 8H, NHCH_2_CH_2_C(O), ^3^*J*_HH_ = 6.8), 2.94–3.03 (m, 8H, CH_2_NHC(O)CH_2_CH_2_), 3.04–3.12 (m, 8H, OCH_2_C(O)NHCH_2_), 3.41 (t, 8H, NHCH_2_CH_2_OH, ^3^*J*_HH_ = 5.7), 3.94 (s, 8H, OCH_2_C(O)), 7.52 (s, 8H, ArH), 7.69 (br.t, 4H, OCH_2_CONH), 7.90 (br.t, 4H, NHC(O)CH_2_CH_2_).

^13^C{^1^H} NMR (DMSO-*d*_6_, δ, ppm): 26.19, 26.32, 29.13, 30.76, 33.87, 35.95, 38.30, 38.74, 45.70, 51.48, 60.31, 70.43, 127.48, 132.11, 146.23, 156.54, 166.89, 171.13.

FTIR ATR (ν, cm^−1^): 3300 (N-H), 1647 (C(O)NH, amide I), 1536 (C(O)NH, amide II), 1085 (C_Ph_OCH_2_).

ESI-HRMS, Calculated [M + 2H]^2+^ *m*/*z* = 904.0099, [M + 3H]^3+^ *m*/*z* = 603.0090. Found [M + 2H]^2+^ *m*/*z* = 904.0129, [M + 3H]^3+^ *m*/*z* = 603.0113.

### 2.5. General Procedure for the Synthesis of Compounds ***16***–***18***

Diethanolamine (0.38 mL, 3.84 mmol) was added to the solution of **4**–**6** (0.20 g, 0.128 mmol) in methanol (10 mL) in glass cylindrical pressure vessel equipped with magnetic stirrer. The reaction mixture was stirred at 100 °C for 40 h. Afterward, the solvent was evaporated under reduced pressure. Then 15 mL of water was added to the residue, and the resulting suspension was centrifuged. The precipitate was dried under reduced pressure.

#### 2.5.1. 5,11,17,23-Tetra-*tert*-butyl-25,26,27,28-tetrakis[*N*-(6-(*N*-(3-(*N*,*N*-di(2-hydroxyethyl)amino)propanoyl)amino)hexyl)carbamoylmethoxy]-2,8,14,20-tetrathiacalix[4]arene **16** in *cone* conformation. Yield: 0.20 g (78%). White Powder, mp 85 °C

^1^H NMR (CD_3_OD, δ, ppm, *J*/Hz): 1.14 (s, 36H, (CH_3_)_3_C), 1.37 (m, 16H, C(O)NHCH_2_CH_2_CH_2_CH_2_), 1.52 (m, 8H, CH_2_CH_2_CH_2_NH), 1.60 (m, 8H, C(O)NHCH_2_CH_2_CH_2_), 2.35 (t, 8H, NHCH_2_CH_2_C(O), ^3^*J*_HH_ = 6.7), 2.65 (t, 16H, N(CH_2_CH_2_OH)_2_, ^3^*J*_HH_ = 5.7), 2.83 (t, 8H, NHCH_2_CH_2_C(O), ^3^*J*_HH_ = 6.7), 3.17 (m, 8H, CH_2_NHC(O)CH_2_CH_2_), 3.33 (m, 8H, OCH_2_C(O)NHCH_2_), 3.60 (t, 16H, N(CH_2_CH_2_OH)_2_, ^3^*J*_HH_ = 5.7), 4.89 (s, 8H, OCH_2_C(O)), 7.44 (s, 8H, ArH).

^13^C{^1^H} NMR (CD_3_OD, δ, ppm): 27.76, 27.78, 30.34, 30.54, 31.63, 31.66, 34.70, 35.20, 40.29, 52.11, 57.24, 60.70, 75.23, 129.77, 136.00, 148.79, 159.20, 170.57, 174.97.

FTIR ATR (ν, cm^−1^): 3300 (N-H), 1648 (C(O)NH, amide I), 1546 (C(O)NH, amide II), 1094 (C_Ph_OCH_2_).

ESI-HRMS, Calculated [M + 2H]^2+^ *m*/*z* = 992.0623, [M + 3H]^3+^ *m*/*z* = 661.7107, [M + 4H]^4+^ *m*/*z* = 496.5348. Found [M + 2H]^2+^ *m*/*z* = 992.0648, [M + 3H]^3+^ *m*/*z* = 661.7132, [M + 4H]^4+^ *m*/*z* = 496.5368.

#### 2.5.2. 5,11,17,23-Tetra-*tert*-butyl-25,26,27,28-tetrakis[*N*-(6-(*N*-(3-(*N*,*N*-di(2-hydroxyethyl)amino)propanoyl)amino)hexyl)carbamoylmethoxy]-2,8,14,20-tetrathiacalix[4]arene **17** in *partial cone* conformation. Yield: 0.20 g (79%). White Powder, mp 81 °C

^1^H NMR (CD_3_OD, δ, ppm, *J*/Hz): 1.08 (s, 18H, (CH_3_)_3_C), 1.20–1.72 (m, 32H, C(O)NHCH_2_CH_2_CH_2_CH_2_, CH_2_CH_2_CH_2_NH, C(O)NHCH_2_CH_2_CH_2_), 1.08 (s, 18H, (CH_3_)_3_C), 2.34 (br.t, 8H, NHCH_2_CH_2_C(O)), 2.65 (br.t, 16H, N(CH_2_CH_2_OH)_2_), 2.83 (t, 8H, NHCH_2_CH_2_C(O)), 3.10–3.25 (m, 16H, CH_2_NHC(O)CH_2_CH_2_, OCH_2_C(O)NHCH_2_), 3.60 (br.t, 16H, N(CH_2_CH_2_OH)_2_), 4.21 (d, 2H, OCH_2_C(O), ^2^*J*_HH_ = 13.8), 4.60 (s, 2H, OCH_2_C(O)), 4.86 (s, 2H, OCH_2_C(O)), 5.02 (d, 2H, OCH_2_C(O), ^2^*J*_HH_ = 13.8), 7.11 (br.d, 2H, ArH), 7.60 (br.d, 2H, ArH), 7.65 (s, 2H, ArH), 7.83 (s, 2H, ArH).

^13^C{^1^H} NMR (CD_3_OD, δ, ppm): 27.71, 27.78, 27.84, 30.37, 30.47, 30.59, 31.68, 31.76, 31.85, 34.71, 35.20, 35.29, 39.94, 40.30, 40.37, 52.13, 57.26, 60.72, 70.59, 74.00, 74.53, 128.03, 129.45, 130.07, 134.67, 135.45, 135.94, 137.50, 147.11, 147.63, 148.87, 157.81, 159.38, 160.83, 170.20, 170.91, 175.06.

FTIR ATR (ν, cm^−1^): 3290 (N-H), 3083 (N-H), 1646 (C(O)NH, amide I), 1542 (C(O)NH, amide II), 1085 (C_Ph_OCH_2_).

ESI-HRMS, Calculated [M + 2H]^2+^ *m*/*z* = 992.0623, [M + 3H]^3+^ *m*/*z* = 661.7107, [M + 4H]^4+^ *m*/*z* = 496.5348. Found [M + 2H]^2+^ *m*/*z* = 992.0648, [M + 3H]^3+^ *m*/*z* = 661.7134, [M + 4H]^4+^ *m*/*z* = 496.5369.

#### 2.5.3. 5,11,17,23-Tetra-*tert*-butyl-25,26,27,28-tetrakis[*N*-(6-(*N*-(3-(*N*,*N*-di(2-hydroxyethyl)amino)propanoyl)amino)hexyl)carbamoylmethoxy]-2,8,14,20-tetrathiacalix[4]arene **18** in *1*,*3-alternate* conformation. Yield: 0.20 g (79%). White Powder, mp 92 °C

^1^H NMR (CD_3_OD, δ, ppm, *J*/Hz): 1.28 (s, 36H, (CH_3_)_3_C), 1.34 (m, 16H, C(O)NHCH_2_CH_2_CH_2_CH_2_), 1.47–1.60 (m, 16H, CH_2_CH_2_CH_2_NH, C(O)NHCH_2_CH_2_CH_2_), 2.35 (t, 8H, NHCH_2_CH_2_C(O), ^3^*J*_HH_ = 6.8), 2.65 (t, 16H, N(CH_2_CH_2_OH)_2_, ^3^*J*_HH_ = 5.7), 2.84 (t, 8H, NHCH_2_CH_2_C(O), ^3^*J*_HH_ = 6.8), 3.12–3.23 (m, 16H, CH_2_NHC(O)CH_2_CH_2_, OCH_2_C(O)NHCH_2_), 3.61 (t, 16H, N(CH_2_CH_2_OH)_2_, ^3^*J*_HH_ = 5.7), 4.16 (s, 8H, OCH_2_C(O)), 7.59 (s, 8H, ArH).

^13^C{^1^H} NMR (CD_3_OD, δ, ppm): 26.65, 26.79, 29.55, 31.21, 33.92, 34.33, 38.80, 39.21, 51.30, 56.64, 59.65, 70.88, 127.93, 132.59, 146.70, 157.00, 167.36, 171.77.

FTIR ATR (ν, cm^−1^): 3300 (N-H), 3074 (N-H), 1646 (C(O)NH, amide I), 1539 (C(O)NH, amide II), 1265 (C(O)NH, amide III), 1027 (C_Ph_OCH_2_).

ESI-HRMS, Calculated [M + 3H]^3+^ *m*/*z* = 661.7107, [M + 4H]^4+^ *m*/*z* = 496.5348. Found [M + 3H]^3+^ *m*/*z* = 661.7132, [M + 4H]^4+^ *m*/*z* = 496.5368.

### 2.6. Procedure for the Synthesis of Compound ***20***

The solution of acryloyl chloride (0.16 mL, 1.94 mmol) in 3 mL of CH_2_Cl_2_ was added dropwise to an ice-cooled (0 °C) mixture of **19** (0.54 g, 1.76 mmol) and triethylamine (0.49 mL, 3.52 mmol) in 5 mL of CH_2_Cl_2_. The reaction mixture was stirred for 3 h at room temperature. Afterward, the reaction mixture was washed with water (5 × 10 mL). Then the organic layer was separated and dried under the anhydrous Na_2_SO_4_. The solvent was removed on a rotary evaporator, and the residue was dried under reduced pressure.

#### *N*-(6-(2-(4-(*tert*-butyl)phenoxy)acetamido)hexyl)acrylamide **20**. Yield: 0.58 g (91%). White Powder, mp 77 °C

^1^H NMR (CDCl_3_, δ, ppm, *J*/Hz): 1.29 (s, 9H, (CH_3_)_3_C), 1.35 (m, 4H, C(O)NHCH_2_CH_2_CH_2_CH_2_), 1.54 (m, 4H, CH_2_CH_2_CH_2_NH, C(O)NHCH_2_CH_2_CH_2_), 3.33 (m, 4H, OCH_2_C(O)NHCH_2_, CH_2_NHC(O)CH=CH_2_), 4.46 (s, 2H, OCH_2_C(O)), 5.62 (dd, 1H, part of ABX system, CH=CH_2_, ^2^*J*_HH_ = 1.6, ^3^*J*_HH_ = 10.2), 5.82 (br.t, 1H, NHC(O)CH=CH_2_), 6.10 (dd, 1H, part of ABX system, CH=CH_2_, ^3^*J*_HH_ = 10.2, ^3^*J*_HH_ = 17.0), 6.28 (dd, 1H, part of ABX system, CH=CH_2_, ^2^*J*_HH_ = 1.6, ^3^*J*_HH_ = 17.0), 6.65 (br.t, 1H, OCH_2_CONH), 6.85 (m, 2H, ArH), 7.33 (m, 2H, ArH).

^13^C{^1^H} NMR (CDCl_3_, δ, ppm): 26.18, 26.23, 29.44, 29.56, 31.58, 34.30, 38.72, 39.29, 67.60, 114.27, 126.35, 126.70, 131.07, 145.06, 155.10, 165.69, 168.60.

FTIR ATR (ν, cm^−1^): 3277 (N-H), 3094 (N-H), 1651 (C(O)NH, amide I), 1624 (C=C), 1513 (C(O)NH, amide II), 1094 (C_Ph_OCH_2_).

### 2.7. General Procedure for the Synthesis of Compounds ***21*** and ***22***

The corresponding diamine (3.10 mmol) (ethylenediamine for **21** and *N*,*N*-dimethylethylenediamine for **22**) was added to the solution of **20** (0.14 g, 0.39 mmol) in 6 mL of methanol. The reaction mixture was refluxed for 15 h. Then the solvent was evaporated under reduced pressure, and the remaining diamine was removed via azeotropic distillation (toluene:methanol mixture, 9:1). Afterward, the remaining toluene was removed via azeotropic distillation with methanol. The residue was dried under reduced pressure.

#### 2.7.1. 3-((2-aminoethyl)amino)-*N*-(6-(2-(4-(*tert*-butyl)phenoxy)acetamido)hexyl)propanamide **21**. Yield: 0.16 g (98%). Viscous Oil

^1^H NMR (CDCl_3_, δ, ppm, *J*/Hz): 1.29 (s, 9H, (CH_3_)_3_C), 1.33 (m, 4H, C(O)NHCH_2_CH_2_CH_2_CH_2_), 1.48 (m, 2H, CH_2_CH_2_CH_2_NH), 1.54 (m, 2H, C(O)NHCH_2_CH_2_CH_2_), 2.37 (t, 2H, NHCH_2_CH_2_C(O), ^3^*J*_HH_ = 5.9), 2.69 (t, 2H, NH_2_CH_2_CH_2_NH, ^3^*J*_HH_ = 5.8), 2.83 (br.t, 2H, NH_2_CH_2_CH_2_NH), 2.89 (t, 2H, NHCH_2_CH_2_C(O), ^3^*J*_HH_ = 5.9), 3.22 (m, 2H, CH_2_NHC(O)CH_2_CH_2_), 3.32 (m, 2H, OCH_2_C(O)NHCH_2_), 4.46 (s, 2H, OCH_2_C(O)), 6.66 (br.t, 1H, OCH_2_CONH), 6.85 (m, 2H, ArH), 7.32 (m, 2H, ArH), 7.48 (br.t, 1H, NHC(O)CH_2_CH_2_).

^13^C{^1^H} NMR (CDCl_3_, δ, ppm): 26.33, 26.45, 29.42, 29.54, 31.58, 34.29, 35.22, 38.86, 39.15, 40.49, 45.28, 50.25, 67.58, 114.28, 126.69, 145.04, 155.09, 168.63, 172.43.

FTIR ATR (ν, cm^−1^): 3289 (N-H), 3065 (N-H), 1647 (C(O)NH, amide I), 1512 (C(O)NH, amide II), 1060 (C_Ph_OCH_2_).

#### 2.7.2. *N*-(6-(2-(4-(*tert*-butyl)phenoxy)acetamido)hexyl)-3-((2-(dimethylamino)ethyl)amino)propanamide **22**. Yield: 0.13 g (75%). Viscous Oil

^1^H NMR (CDCl_3_, δ, ppm, *J*/Hz): 1.29 (s, 9H, (CH_3_)_3_C), 1.33 (m, 4H, C(O)NHCH_2_CH_2_CH_2_CH_2_), 1.48 (m, 2H, CH_2_CH_2_CH_2_NH), 1.54 (m, 2H, C(O)NHCH_2_CH_2_CH_2_), 2.22 (s, 6H, N(CH_3_)_2_), 2.36 (t, 2H, NHCH_2_CH_2_C(O), ^3^*J*_HH_ = 5.9), 2.41 (t, 2H, (CH_3_)_2_NCH_2_CH_2_NH, ^3^*J*_HH_ = 6.0), 2.70 (br.t, 2H, (CH_3_)_2_CH_2_CH_2_NH), 2.88 (br.t, 2H, NHCH_2_CH_2_C(O)), 3.20 (m, 2H, CH_2_NHC(O)CH_2_CH_2_), 3.32 (m, 2H, OCH_2_C(O)NHCH_2_), 4.46 (s, 2H, OCH_2_C(O)), 6.63 (br.t, 1H, OCH_2_CONH), 6.85 (m, 2H, ArH), 7.32 (m, 2H, ArH), 7.75 (br.t, 1H, NHC(O)CH_2_CH_2_).

^13^C{^1^H} NMR (CDCl_3_, δ, ppm): 26.45, 26.56, 29.51, 29.58, 31.57, 34.28, 35.35, 38.90, 39.01, 45.55, 45.69, 46.66, 58.53, 67.59, 114.26, 126.67, 145.02, 155.09, 168.50, 172.62.

FTIR ATR (ν, cm^−1^): 3300 (N-H), 3065 (N-H), 1649 (C(O)NH, amide I), 1512 (C(O)NH, amide II), 1056 (C_Ph_OCH_2_).

### 2.8. General Procedure for the Synthesis of Compounds ***23*** and ***24***

The corresponding alkanolamine (3.10 mmol) (ethanolamine for **23** and diethanolamine for **24**) was added to the solution of **20** (0.14 g, 0.39 mmol) in 6 mL of methanol. The reaction mixture was refluxed for 15 h. Then the solvent was evaporated under reduced pressure. Then, 5 mL of water was added to the residue, and the resulting emulsion was centrifuged. The precipitate was dried under reduced pressure.

#### 2.8.1. *N*-(6-(2-(4-(*tert*-butyl)phenoxy)acetamido)hexyl)-3-((2-hydroxyethyl)amino)propanamide **23**. Yield: 0.08 g (49%). Viscous Oil

^1^H NMR (CDCl_3_, δ, ppm, *J*/Hz): 1.29 (s, 9H, (CH_3_)_3_C), 1.34 (m, 4H, C(O)NHCH_2_CH_2_CH_2_CH_2_), 1.49 (m, 2H, CH_2_CH_2_CH_2_NH), 1.55 (m, 2H, C(O)NHCH_2_CH_2_CH_2_), 2.39 (t, 2H, NHCH_2_CH_2_C(O), ^3^*J*_HH_ = 5.9), 2.80 (t, 2H, NHCH_2_CH_2_OH, ^3^*J*_HH_ = 5.1), 2.92 (t, 2H, NHCH_2_CH_2_C(O), ^3^*J*_HH_ = 5.9), 3.23 (m, 2H, CH_2_NHC(O)CH_2_CH_2_), 3.33 (m, 2H, OCH_2_C(O)NHCH_2_), 3.70 (t, 2H, NHCH_2_CH_2_OH, ^3^*J*_HH_ = 5.1), 4.46 (s, 8H, OCH_2_C(O)), 6.68 (br.t, 1H, OCH_2_CONH), 6.85 (m, 2H, ArH), 7.29 (br.t, 1H, NHC(O)CH_2_CH_2_), 7.32 (m, 2H, ArH).

^13^C{^1^H} NMR (CDCl_3_, δ, ppm): 26.22, 26.33, 29.34, 29.46, 31.58, 34.30, 35.43, 38.78, 39.07, 45.32, 50.94, 60.76, 67.55, 114.27, 126.71, 145.09, 155.06, 168.76, 172.39.

FTIR ATR (ν, cm^−1^): 3307 (N-H), 3065 (N-H), 1633 (C(O)NH, amide I), 1540 (C(O)NH, amide II), 1066 (C_Ph_OCH_2_).

#### 2.8.2. 3-(*Bis*(2-hydroxyethyl)amino)-*N*-(6-(2-(4-(*tert*-butyl)phenoxy)acetamido)hexyl)propanamide **24**. Yield: 0.10 g (56%). Viscous Oil

^1^H NMR (CDCl_3_, δ, ppm, *J*/Hz): 1.28 (s, 9H, (CH_3_)_3_C), 1.32 (m, 4H, C(O)NHCH_2_CH_2_CH_2_CH_2_), 1.45–1.59 (m, 4H, CH_2_CH_2_CH_2_NH, C(O)NHCH_2_CH_2_CH_2_), 2.43 (t, 2H, NHCH_2_CH_2_C(O), ^3^*J*_HH_ = 6.0), 2.75 (t, 4H, N(CH_2_CH_2_OH)_2_, ^3^*J*_HH_ = 5.0), 2.94 (t, 2H, NHCH_2_CH_2_C(O), ^3^*J*_HH_ = 6.0), 3.22 (m, 2H, CH_2_NHC(O)CH_2_CH_2_), 3.31 (m, 2H, OCH_2_C(O)NHCH_2_), 3.68 (t, 4H, N(CH_2_CH_2_OH)_2_, ^3^*J*_HH_ = 5.0), 4.45 (s, 8H, OCH_2_C(O)), 6.79 (br.t, 1H, OCH_2_CONH), 6.84 (m, 2H, ArH), 7.04 (br.t, 1H, NHC(O)CH_2_CH_2_), 7.31 (m, 2H, ArH).

^13^C{^1^H} NMR (CDCl_3_, δ, ppm): 25.96, 26.09, 29.16, 29.32, 31.59, 33.86, 34.31, 38.71, 39.35, 51.24, 56.58, 59.03, 67.53, 114.29, 126.72, 145.09, 155.06, 168.88, 172.21.

FTIR ATR (ν, cm^−1^): 3289 (N-H), 3096 (N-H), 1645 (C(O)NH, amide I), 1512 (C(O)NH, amide II), 1056 (C_Ph_OCH_2_).

## 3. Results and Discussion

### 3.1. Development of an Approach to the Synthesis of Multivalent Derivatives of Thiacalix[4]arene

According to the literature [43,59], most of the previously reported macrocyclic antibacterial compounds based on calix[n]arenes are in *cone* conformation, in which four lower rim substituents are located on one side of the macrocyclic core (Figure 1). It was noted that the macrocycle conformation (and consequently the spatial orientation of the substituents) has a significant influence on its antibacterial activity. Therefore, the synthesis of new (thia)calixarenes in different conformations is a promising task. Its solving will allow us to establish new structure–activity relationships for macrocycle conformations and its antibacterial activity and to find the most effective derivatives of (thia)calixarene with improved antibacterial properties. Therefore, we specifically chose thiacalix[4]arene as a macrocyclic platform because its three conformations (*cone*, *partial cone*, and *1*,*3-alternate*) can be synthesized in high yields (Figure 1). As starting compounds, we chose thiacalix[4]arene derivatives **1**–**3**, previously obtained in our research group [61], containing four amidoamine fragments at the lower rim of the macrocycle. These macrocycles can be easily obtained in high yields in two steps from the starting *p-tert*-butylthiacalix[4]arene in three conformations, i.e., *cone*, *partial cone*, and *1*,*3-alternate*. The presence of terminal primary amino groups in these compounds opens up wide possibilities for further modification of the macrocycle. At the same time, the hexylidene spacer provides sufficient distance between the reaction centers to reduce the possibility of side reactions.

In order to introduce additional amino groups of different nature (i.e., primary, secondary, and tertiary) into the structure of the starting thiacalixarenes, one of the priority tasks was to design of macrocyclic precursors capable of easily forming new C-N bonds without changing the amine nature of the reagent. For this purpose, we have selected an acrylamide fragment that is a Michael acceptor and can be readily involved in the addition reaction of N-nucleophiles to electron-deficient alkenes (aza-Michael reaction) [62,63], hence ideally suited to the above requirements (Figure 1).

The interaction of such macrocyclic precursors with a series of diamines and alkanolamines (ethylenediamine, *N*,*N*-dimethylethylenediamine, ethanolamine, and diethanolamine) will lead to polyfunctional derivatives of thiacalixarene in various conformations (*cone*, *partial cone*, and *1*,*3-alternate*), containing various amine fragments (primary, secondary, and tertiary) and hydroxyl groups, potentially possessing antimicrobial activity (Figure 1). In addition, once the optimal conditions for these reactions are established, it will be possible in the future to introduce the macrocyclic acrylamide precursors obtained into the reaction with more complex reagents containing an N-nucleophilic site.

The first stage of the synthetic work involved the synthesis of precursors of target multivalent thiacalixarene derivatives, i.e., macrocycles containing acrylamide groups. For this purpose, starting macrocycles **1**–**3** were introduced into the acylation reaction with acryloyl chloride. For compounds **1** (*cone*) and **3** (*1*,*3-alternate*), the synthesis was carried out in dichloromethane. In the case of compound **2** in *partial cone* conformation, chloroform was chosen as the solvent due to its poor solubility in dichloromethane. After addition of the acryloyl chloride at low temperature, the reaction mixtures were stirred for 12 h at room temperature. A number of methods, e.g., extraction, recrystallization, and column chromatography, were successively tried for the purification of acrylamide precursors **4**–**6**, but these methods were found to be inefficient. It was found that the most optimal method for purification of these compounds was water re-precipitation from saturated ethanol solution. As a result, macrocyclic precursors **4**–**6** in *cone*, *partial cone*, and *1*,*3-alternate* conformations were obtained in high yields (85–96%).

In the second synthetic step, obtained acrylamide precursors **4**–**6** as Michael acceptors were reacted with a series of amines and alkanolamines (ethylenediamine, *N*,*N*-dimethylethylenediamine, ethanolamine, and diethanolamine). Initially, the reaction of compounds **4**–**6** with ethylenediamine was investigated. To avoid side reactions (e.g., intermolecular and intramolecular cross-linking), a 30-fold excess of the corresponding amine per macrocycle molecule was used. The reaction mixture in methanol was refluxed for 60 h. To remove excess amine from the reaction mixture, a convenient and practical method of azeotropic distillation with toluene:methanol mixture (9:1), which we had previously used in the synthesis of PAMAM-calix-dendrimers, was used [64,65]. As a result, macrocycles **7**–**9** in *cone*, *partial cone*, and *1*,*3-alternate* conformations were isolated with 97%, 92%, and 90% yields, respectively.

In the next step, compounds **4**–**6** were reacted with *N*,*N*-dimethylethylenediamine and ethanolamine containing only one primary amino group. The reactions were carried out under similar conditions, with excess reagent in methanol under reflux. After 60 h, according to the ^1^H NMR spectroscopy data, the reaction was not complete, as indicated by residual signals of acrylamide fragments in the range of 5.55–6.18 ppm in the ^1^H NMR spectra of the reaction mixtures. We therefore decided to extend the reaction time to 90 h. As a result, the target polyfunctional thiacalix[4]arene derivatives **10**–**12** and **13**–**15** were isolated in 94–98% and 79–84% yields, respectively.

The reactions of compounds **4**–**6** with diethanolamine were initially carried out analogously to the previous syntheses. However, even after 170 h of the reflux, the conversion of the reagent into the product was not complete. Apparently, this is due to the increased steric hindrance of the diethanolamine amino group compared to the primary amines used. A glass autoclave was used to increase the conversion and speed up the reaction. This allowed us to significantly reduce the synthesis time and achieve complete conversion (Figure 1). The reaction was carried out at 100 °C for 40 h in methanol. Upon completion of the reaction, macrocycles **16**–**18** were isolated from the reaction mixtures in yields of 78–79%.

In order to evaluate the influence of the macrocyclic platform on the biological activity of the obtained compounds, we have additionally synthesized monomeric analogues of thiacalix[4]arenes **7**–**18**, i.e., *p-tert*-butylphenol derivatives containing identical substituents. For this purpose, acylation of amine **19** with acryloyl chloride was carried out. This resulted in compound **20**, which was involved in the next step in the aza-Michael addition reaction with the above-described series of amines and alkanols (ethylenediamine, *N*,*N*-dimethylethylenediamine, ethanolamine, and diethanolamine). Thus, monomeric compounds, *p-tert*-butylphenol derivatives **21**–**24**, were obtained in 49–98% yields.

All the obtained compounds were fully characterized by a number of physical methods such as ^1^H, ^13^C{^1^H} NMR, IR spectroscopy, and ESI high resolution mass spectrometry (ESI-HRMS) (Appendix A).

Thus, in all the ^1^H NMR (DMSO-*d*_6_) spectra of obtained acrylamide derivatives of thiacalix[4]arene **4**–**6**, the protons at the double bond of the acrylamide fragment were presented as ABX-system expressed as doublets of doublets at 5.55, 6.06, and 6.18 ppm. The protons at the nitrogen atoms of the acrylamide fragments appeared as a broadened triplet at 8.05 ppm. The proton signals of the acrylamide fragment were observed in the same regions of the ^1^H NMR spectra regardless of the macrocycle conformation due to their sufficient distance from the macrocyclic platform. In the ^1^H NMR spectra of synthesized macrocycles **7**–**18**, the signals of the acrylic fragments in the above-mentioned regions were completely absent. Together with the ratio of the signal intensities and the multiplicity of peaks, this confirmed the completion of the reaction. In all ^1^H NMR spectra of compounds **7**–**15**, triplets of two methylene groups of C(O)–CH_2_–CH_2_–R fragments were observed at 2.18 and 2.68 ppm. The signals of the methylene group protons of the terminal ethylidene fragment closest to the secondary amino group appeared as a broadened triplet at 2.53–2.57 ppm. The proton signals of the end methylene group of the terminal ethylidene fragment appeared as broadened triplets at 2.47, 2.25, and 3.40 ppm for macrocycles with terminal primary, tertiary amino or hydroxyl groups, respectively. The thiacalixarene conformation only affected the chemical shifts of the protons closest to the macrocyclic platform. Thus, in the ^1^H NMR spectra (DMSO-*d*_6_) of all the compounds obtained in *cone* conformation, the proton signals of the *tert*-butyl, oxymethylene, and aromatic fragments appeared as singlets at 1.06, 4.75, and 7.38 ppm, respectively. At the same time, the signals of these protons for the compounds in *1*,*3-alternate* conformation were located at 1.19, 3.94, and 7.52 ppm, due to the shielding of these protons by the aromatic rings of the macrocycle in this conformation. All of the above trends are also fully presented in the ^1^H NMR spectra of compounds **16**–**18**, which were recorded in CD_3_OD to simplify the interpretation.

In the IR spectra of acrylamide compounds **4**–**6**, a narrow intense absorption band related to the C=C bond vibrations was observed at 1625 cm^−1^. This characteristic absorption band was completely absent in the IR spectra of compounds **7**–**18**, which further confirmed the structure of the obtained compounds. Along with this, the IR spectra of compounds **4**–**18** contained broad bands at the 3000 and 3075 cm^−1^ corresponding to N-H vibrations, bands at 1650 and 1540 cm^−1^ (amide I and amide II), as well as a band at the 1080–1095 cm^−1^ characteristic for arylalkyl ethers (C_Ph_OCH_2_ fragment).

The obtained compounds were additionally characterized by mass spectrometry (ESI-HRMS). Peaks of mono- or diprotonated molecules [M + 1H]^1+^ and [M + 2H]^2+^ were registered in all mass spectra of acrylamide thiacalixarene derivatives **4**–**6**. In the case of compounds **7**–**18**, the peaks of di- and triprotonated molecules [M + 2H]^2+^ and [M + 3H]^3+^ were observed in all mass spectra. Additionally, peaks of tetraprotonated ions [M + 4H]^4+^ were found in mass spectra of compounds **10**–**12** and **15**–**18** containing the tertiary amino groups.

Thus, a convenient synthetic approach for multivalent thiacalix[4]arene derivatives containing amide, hydroxyl, and amino groups, consisting of a stepwise modification of the macrocyclic platform with acrylic fragments and the subsequent reaction with diamines and alkanolamines has been developed.

### 3.2. Antibacterial Properties and Cytotoxicity of the Obtained Multivalent Derivatives of Thiacalix[4]arene

According to the literature [66,67,68], the mechanism of antibacterial action of most compounds containing amine and ammonium groups is based on interaction with the negatively charged cell membrane of bacteria. Sufficient examples of ammonium and amine derivatives of (thia)calixarenes with antibacterial activity have been reported in the literature [44,69]. However, structure–activity relationship data establishing the antibacterial activity of the macrocycle with its conformation are currently lacking. Moreover, the *partial cone* conformation data are limited to individual examples [52]. The establishment of such structure–activity relationships will allow for the design of macrocyclic antibiotics for specific applications by exploiting the features of each conformation.

It is well known that (thia)calixarene derivatives in *1*,*3-alternate* stereoisomeric form containing long substituents with polar terminal fragments are able to incorporate into lipid bilayers via “bouquet”-type (Figure 2), forming ion channel-type structures [70,71]. In this case, the lipophilic macrocyclic part of the molecule is incorporated in the middle of the phospholipid membrane, and substituents with polar terminal fragments are built at the boundaries of the membrane bilayer next to the polar groups. Such incorporation causes loosening of the lipid bilayer, leading to the destruction of bacterial cell membranes. This “bouquet”-type incorporation into membranes can also be expected in the case of macrocycles in *partial cone* conformation. However, prior to this work, there are no literature data describing the antibacterial activity of (thia)calixarene derivatives in *partial cone* conformation relative to *1*,*3-alternate*.

To establish the regularities between the structure of the synthesized compounds and their biological activity, the antibacterial activity of obtained macrocyclic compounds **7**–**18** and monomeric analogues **21**–**24** against Gram-positive (*Staphylococcus aureus*, *Bacillus cereus*, and *Enterococcus faecalis*) and Gram-negative (*Escherichia coli* and *Pseudomonas aeruginosa*) bacterial strains was further studied. The well-known antibiotics Ciprofloxacin and Norfloxacin were chosen as the standards. The dilutions of the compounds were prepared immediately in nutrient media; 5% DMSO was added for better solubility and the test strains were not inhibited at this concentration. The investigated macrocyclic compounds were found to have high antibacterial activity against the studied bacterial strains, while the monomeric compounds were characterized by low activity (Table 1). These results clearly indicate the advantage of the multivalent approach for the development of antibacterial agents. It should be noted that thiacalixarene derivatives in *partial cone* conformation had better antibacterial activity compared to the compounds in *cone* and *1*,*3-alternate* stereoisomeric forms. In our opinion, this is due to the fact that *partial cone* is the most dissymmetric in *cone*-*partial cone*-*1*,*3-alternate* conformer series. Therefore, compounds in *partial cone* conformation tend to loosen membrane bilayers most strongly upon incorporation (Figure 2). This seems to be the reason for the highest activity of compounds in *partial cone* conformation. In turn, compounds in *1*,*3-alternate* conformation are symmetric and capable of tighter packing when incorporated into lipid bilayers, resulting in less membrane loosening in bacteria. Therefore, the antibacterial activity of macrocycle compounds in *1*,*3-alternate* conformation was lower. At the same time, compounds in *cone* conformation are not able to incorporate into membranes in the “bouquet”-type manner described above due to their amphiphilic structure in which the lipophilic *tert*-butyl groups and the hydrophilic substituents of the lower rim are on opposite sides of the macrocyclic platform. For incorporation into a membrane, a pair of macrocycles in *cone* conformation must be oriented toward each other with lipophilic parts (Figure 2). Thus, two molecules of the *cone* macrocycle are required to overlap the membrane bilayer, corresponding to the “slot” type of structures [40]. This is an important difference from the macrocycles obtained in *1*,*3-alternate* and *partial cone* conformations, in the case of which a single macrocycle molecule is required to overlap the bilayer.

Among macrocycles with different substituents, compounds **10**–**12** containing -NHCH_2_CH_2_N(CH_3_)_2_ fragments had the highest antibacterial activity against all studied bacterial strains. These results are explained by their greater lipophilicity—the calculated miLogP values for these compounds were the highest (Table 1, miLogP values were calculated using an online platform http://www.molinspiration.com/cgi-bin/properties (accessed on 10 October 2023). The calculated values of MW (molecular weight), miLogP (logarithm of the octanol–water partition coefficient), HBA (hydrogen bond acceptor atoms), HBD (hydrogen bond donor atoms), TPSA (topological polar surface area), and solubility data are also presented in Appendix A. The introduction of additional terminal hydroxyl groups (when comparing compounds **13**–**15** with -NHCH_2_CH_2_OH fragments and compounds **16**–**18** with -N(CH_2_CH_2_CH_2_OH)_2_ fragments) resulted in a dramatic decrease in antibacterial activity against all bacterial strains studied. This can be explained by the reduction of surface positive charge in thiacalixarene derivatives **16**–**18** upon introduction of additional terminal hydroxyl groups.

An important characteristic in the development of new drugs is their cytotoxic effect on mammalian cells. Therefore, the next step of this work was to study the cytotoxicity of lead compounds **10**–**12** against normal Chang liver cell line (Human liver cells) (Table 2). The advantage of in vitro models is the ability to work directly on human cell cultures, which makes the data obtained more adequate when applied to the human body. In addition, the use of cell cultures makes it possible to establish the nature of the biological activity of the studied compounds directly at the cellular level and take into account the complex synergistic or multidirectional effects of mixtures of chemical compounds [72]. The lowest cytotoxicity value was observed for macrocycle **11** in *partial cone* conformation (IC_50_ = 3.6 ± 0.3 M). This fact also confirmed the previously proposed mechanism of incorporation of the obtained compounds into the membrane bilayer. The cytotoxicity values for macrocycles **10** (IC_50_ = 52.0 ± 4.2 M) in *cone* conformation and **12** in *1*,*3-alternate* conformation (IC_50_ = 25.3 ± 1.8 M) were concentrationally consistent (2-fold difference) with the mechanism proposed above, according to which two macrocycle molecules are involved in bilayer overlapping in the case of compounds in *cone* conformation. Compound **11** (*partial cone*) with the most significant antibacterial activity was predominantly non-selective against cells (highest selective index (SI) = 4.0 for *S. aureus*). Compound **10** (*cone*) showed the highest selectivity against *S. aureus* (SI = 3.3) and *E. faecalis* (SI = 13.3).

Thus, the compounds in *partial cone* conformation had significant bactericidal activity against all studied cell strains. This can be used to create drugs with universal action against all microorganisms. At the same time, the best selectivity was observed for the compounds in *cone* conformation, which can be used to design antibacterial agents with low toxicity and activity against a specific type of bacteria. Certainly, this work is a proof of concept, and further optimization of the structure–activity relationship is required. However, we already believe that the proposed universal approach to the synthesis of multivalent antimicrobial agents has great potential for the future.

### 3.3. Study of the Mechanism of Antibacterial Activity of the Obtained Compounds

The bacterial membrane mimetic systems were further used to prove the proposed mechanism of interaction of the compounds obtained with bacteria. Lead compounds **10**–**12** in *cone*, *partial cone*, and *1*,*3-altenate* conformations, were chosen for this experiment. Initially, the self-association of compounds **10**–**12** was investigated by dynamic light scattering (DLS) in TRIS buffer (pH = 7.4, 150 mM NaCl) at concentration 1 × 10^−5^ M and 1 × 10^−4^ M. Typically, supramolecular systems with low polydispersity index (PDI < 0.25) values are considered stable and monodisperse, but no formation of stable supramolecular systems based on compounds **10**–**12** (PDI = 0.32–0.70) was found (Appendix A). Consequently, the antibacterial activity of the obtained compounds was due to the action of the macrocycles themselves rather than their self-associates.

Model vesicles based on 1-palmitoyl-2-oleoyl-*sn*-glycero-3-phospho-(1′-*rac*-glycerol) sodium salt (POPG) were used as Gram-positive negatively charged membrane mimetic systems [73,74,75]. POPG vesicles were prepared in TRIS buffer (pH = 7.4, 150 mM NaCl). The resulting vesicles were studied by DLS and Doppler microelectrophoresis to evaluate the effect of macrocycles addition (Table 3). The values of the hydrodynamic diameter and the electrokinetic potential (zeta potential) of the vesicles were measured in presence and absence of compounds **10**–**12**. Measurements were carried out in molar ratios [POPG:macrocycle] = 1:0.1 and [POPG:macrocycle] = 1:1. As the concentration of macrocycles **10**–**12** increased, the hydrodynamic diameter of the resulting vesicles increased. The most noticeable changes occurred in the case of **12** (*1*,*3-altertane*) (up to 290 nm) and **11** (*partial cone*) (up to 571 nm). Moreover, in the case of **11** (*partial cone*), the monodispersity of the POPG vesicles was disrupted (PDI 0.56) and the hydrodynamic diameter was very different from the initial one. This can be explained by partial destruction of the POPG vesicles. The zeta potential values also increased with increasing macrocycle concentration, which clearly indicated the adsorption of the compounds on the surface of lipid vesicles and further incorporation into the membrane structure.

The obtained data additionally confirmed the previously proposed mechanism of interaction of the obtained compounds with the membrane bilayer. The largest changes in the size of model vesicles (1:1 ratio) were observed in the case of macrocycle **11** (*partial cone*) since the compounds in this conformation loosen the bilayer the most during incorporation. Smaller changes were found for compound **12** (*1*,*3-alternate*). In the case of macrocycle **10** (*cone*), POPG vesicle size changes were the smallest. This is due to the fact that twice as many macrocycle molecules in *cone* conformation, as opposed to *partial cone* and *1*,*3-alternate*, are required to overlap the bilayer upon incorporation. Thus, the concentration of compound **10** used is not sufficient to induce significant changes in POPG vesicle size.

Thus, the mechanism of antibacterial activity of lead compounds **10**–**12** was proved using model negatively charged POPG vesicles. The addition of **12** (*partial cone*), which had the best biological activity, was also found to result in partial destruction of the vesicles.

## 4. Conclusions

In this work, a synthetic approach to obtain multivalent thiacalix[4]arene derivatives containing different amounts of amine and hydroxyl groups was developed for the first time. A series of macrocyclic compounds in *cone*, *partial cone*, and *1*,*3-alternate* stereoisomeric forms containing -NHCH_2_CH_2_R (R = NH_2_, N(CH_3_)_2_, and OH) and -N(CH_2_CH_2_OH)_2_ fragments and their model non-macrocyclic analogs were prepared. The antibacterial activity against Gram-positive (*Staphylococcus aureus*, *Bacillus cereus*, and *Enterococcus faecalis*) and Gram-negative (*Escherichia coli* and *Pseudomonas aeruginosa*) bacterial strains and cytotoxicity of the obtained compounds were studied, and structure–activity relationships were established. The synthesized macrocyclic compounds were found to have better biological activity compared to monomeric analogs. The mechanism of antibacterial action of the obtained compounds lies in the interaction with negatively charged cell membrane of bacteria, which was proved on the example of model vesicles POPG. The compounds in *partial cone* conformation had significant bactericidal activity against all studied cell strains. The best selectivity was observed for the compounds in *cone* conformation.

Certainly, this work is a proof of concept, and further optimization of the structure–activity relationship is required. However, we already believe that the proposed universal approach to the synthesis of multivalent antimicrobial agents has great potential for the future. The obtained results open up the possibility of using the synthesized macrocyclic compounds as promising antibacterial agents.

## Data Availability

The data presented in this study are available in the Appendix A.

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
