# Peer review of "Towards Antibacterial Agents: Synthesis and Biological Activity of Multivalent Amide Derivatives of Thiacalix[4]arene with Hydroxyl and Amine Groups"

_pharmaceutics, 2023, doi:10.3390/pharmaceutics15122731_

Round 1
Reviewer 1 Report
Comments and Suggestions for Authors
the manuscript needs extensive changes.
Please write again the general “experimental information” section indicating exactly what is reported in supplementary material. It is not clear to the reader what is going to be found in the supplementary section and consequently it is very hard to correlate reported experiments to a technique.
In the table1 authors need to show results also in presence of DMSO.
Comments on the Quality of English Languagesmall changes are required
line 584 and 588 needs to be redited
Author Response
Reviewer #1:
the manuscript needs extensive changes.
Please write again the general “experimental information” section indicating exactly what is reported in supplementary material. It is not clear to the reader what is going to be found in the supplementary section and consequently it is very hard to correlate reported experiments to a technique.
Response:
Dear Reviewer! Thank you very much for carefully reading and reviewing our paper! We are very pleased that you have given it such a high rating.
More details on the experimental information from the supplementary materials have been added to the manuscript.
In the table1 authors need to show results also in presence of DMSO.
Response:
The dilutions of the compounds were prepared immediately in nutrient media; 5% DMSO was added for better solubility and the test strains were not inhibited at this concentration. These data were added to the manuscript.

Reviewer 2 Report
Comments and Suggestions for Authors
In the presented manuscript, the Authors synthesized thiacalix[4]arene derivatives with different numbers of amine and hydroxyl groups and tested their antimicrobial and cytotoxic activities. The mode of action of selected compounds was demonstrated using POPG vesicles mimicking bacterial membranes. The presented results are interesting and convincing.
In the electronic supplementary information, point 2.1., the Authors should add information on how the Minimum inhibitory concentration was estimated.
Due to differences in the structure of the cell envelope, can the mechanism of action of these compounds be different in the case of gram-positive and gram-negative bacteria?
Author Response
In the presented manuscript, the Authors synthesized thiacalix[4]arene derivatives with different numbers of amine and hydroxyl groups and tested their antimicrobial and cytotoxic activities. The mode of action of selected compounds was demonstrated using POPG vesicles mimicking bacterial membranes. The presented results are interesting and convincing.
Response:
Dear Reviewer! Thank you very much for carefully reading and reviewing our paper! We are very pleased that you have given it such a high rating.
In the electronic supplementary information, point 2.1., the Authors should add information on how the Minimum inhibitory concentration was estimated.
Response:
The information on the minimum inhibitory concentration (MIC) estimation has been added to the supplementary materials:
"The minimum inhibitory concentration (MIC) was defined as the minimum concentration of a compound that inhibits the growth of the corresponding test microorganism. The growth of bacteria as well as the absence of the growth due to the bacteriostatic action of a compounds were recorded."
Due to differences in the structure of the cell envelope, can the mechanism of action of these compounds be different in the case of gram-positive and gram-negative bacteria?
Response:
It is known that Gram-positive and Gram-negative bacteria have different cell wall structures and respond differently to external influences. This is primarily due to the fact that Gram-negative bacteria have an additional outer membrane that makes it difficult for compounds to penetrate the cell, which increases their resistance to antimicrobial agents, including surfactants, while the single-layer membranes of Gram-positive bacteria are much more sensitive to them [Salton MRJ, Kim KS. Structure. In: Baron S, editor. Medical Microbiology. 4th edition. Galveston (TX): University of Texas Medical Branch at Galveston; 1996. Chapter 2.].

Reviewer 3 Report
Comments and Suggestions for Authors
The manuscript is well structured, written, and easy to read, even by less experienced readers. From the perspective of discovering and developing new antibacterials, the work is valuable and deserves to be published. However, to my knowledge, no calixa[4]arene has been approved for therapy for any biological effect.
1. Perhaps it is helpful to find out the potential targets on which these compounds act (see the programs available for predictions) and verify those possible biological effects and their predicted toxicity. Also, there are several computational methods to find more specific data regarding the predictions of the physicochemical properties of these compounds, which is very useful to determine the drug-likeness of this series. It would be helpful to introduce a synthetic table with the properties of the obtained compounds (MW, HBD, HBA..., solubility, pka, etc.).
2. It would have been helpful to determine antibacterial activity by choosing an additional standard with a specific activity on Gram-positive bacteria. Thus, the antibacterial activity of the synthesised compounds does not exceed that of ciprofloxacin (known for activity especially on Gram-negative pathogens). However, it is possible that the synthesised compounds are much more active on certain bacterial species than classic antibacterials. The trend in developing new antibiotics is identifying antibacterials with a narrow spectrum.
3. The determination of the cytotoxicity of the compounds is very briefly addressed and argued.
4. Minor comment: Line 38: Explain the abbreviation of WHO at the first use in the manuscript
Author Response
The manuscript is well structured, written, and easy to read, even by less experienced readers. From the perspective of discovering and developing new antibacterials, the work is valuable and deserves to be published. However, to my knowledge, no calixa[4]arene has been approved for therapy for any biological effect.
Response:
Dear Reviewer! Thank you very much for carefully reading and reviewing our paper! We are very pleased that you have given it such a high rating.
1 Perhaps it is helpful to find out the potential targets on which these compounds act (see the programs available for predictions) and verify those possible biological effects and their predicted toxicity. Also, there are several computational methods to find more specific data regarding the predictions of the physicochemical properties of these compounds, which is very useful to determine the drug-likeness of this series. It would be helpful to introduce a synthetic table with the properties of the obtained compounds (MW, HBD, HBA..., solubility, pka, etc.).
Response:
Unfortunately, the obtained macrocyclic compounds have large molecular weight, so it is impossible to predict their physicochemical parameters and biological activity using computational methods. Table S3 with the properties of the obtained compounds has been added in the supplementary materials.
- It would have been helpful to determine antibacterial activity by choosing an additional standard with a specific activity on Gram-positive bacteria. Thus, the antibacterial activity of the synthesised compounds does not exceed that of ciprofloxacin (known for activity especially on Gram-negative pathogens). However, it is possible that the synthesised compounds are much more active on certain bacterial species than classic antibacterials. The trend in developing new antibiotics is identifying antibacterials with a narrow spectrum.
Response:
We have added the MIC and MBC data of the additional antibiotic Norfloxacin to Table 1. The lead synthesized macrocyclic compounds had biological activity values at the level of Norfloxacin.
- The determination of the cytotoxicity of the compounds is very briefly addressed and argued.
Response:
The part on determination of cytotoxicity of the synthesized compounds was expanded. The discussion was added.
- Minor comment: Line 38: Explain the abbreviation of WHO at the first use in the manuscript
Response:
Full name of the World Health Organization has been added to the manuscript.

Round 2
Reviewer 1 Report
Comments and Suggestions for Authors
The paper is acceptable